# Substrate evaporation drives collective construction in termites

**Giulio Facchini[1,2,3]\*, Alann Rathery[1], Stéphane Douady[3], David Sillam-Dussès[4], Andrea Perna[1,5]**

[1]Life Sciences Department, University of Roehampton, London, United Kingdom; [2]Service de Chimie et Physique Non Linéaire, Université Libre de Bruxelles, Brussels, Belgium; [3]Laboratoire Matière et Systèmes Complexe, CNRS, Université Paris Cité, Paris, France; [4]Laboratoire d'Ethologie Expérimentale et Comparée, LEEC, UR 4443, Université Sorbonne Paris Nord, Villetaneuse, France; [5]Networks Unit, IMT School for Advanced Studies Lucca, Lucca, Italy

**Abstract** Termites build complex nests which are an impressive example of self-organization. We know that the coordinated actions involved in the construction of these nests by multiple individuals are primarily mediated by signals and cues embedded in the structure of the nest itself. However, to date there is still no scientific consensus about the nature of the stimuli that guide termite construction, and how they are sensed by termites. In order to address these questions, we studied the early building behavior of *Coptotermes gestroi* termites in artificial arenas, decorated with topographic cues to stimulate construction. Pellet collections were evenly distributed across the experimental setup, compatible with a collection mechanism that is not affected by local topography, but only by the distribution of termite occupancy (termites pick pellets at the positions where they are). Conversely, pellet depositions were concentrated at locations of high surface curvature and at the boundaries between different types of substrate. The single feature shared by all pellet deposition regions was that they correspond to local maxima in the evaporation flux. We can show analytically and we confirm experimentally that evaporation flux is directly proportional to the local curvature of nest surfaces. Taken together, our results indicate that surface curvature is sufficient to organize termite building activity and that termites likely sense curvature indirectly through substrate evaporation. Our findings reconcile the apparently discordant results of previous studies.

**\*For correspondence:**
giuliofacchini@gmail.com

**Competing interest:** The authors declare that no competing interests exist.

## eLife assessment

This **valuable** study investigates the environmental drivers behind termite construction, focusing, in particular, on pellet deposition behavior, with the conclusion that termites likely sense curvature indirectly through substrate evaporation. The findings reconcile discrepancies between previous studies through experimental and computational approaches. While the strength of the evidence supporting these claims is **compelling**, the authors do not discuss how their results affect our understanding of insect nest construction or animal-built structures more broadly.

## Introduction

Termites are known for their ability to build some of the most complex nests and shelters found in nature (*Hansell, 2005*; *Perna and Theraulaz, 2017*). The construction of these structures is achieved through the collective actions of multiple individual workers (up to thousands or millions in large termite colonies) each performing the collection, transportation and deposition of elementary pellets. In order to produce functionally meaningful structures, it is essential that all these different workers

operate in a coordinated, coherent way, each continuing the work started by their colony mates, rather than undoing it.

Termites rely on individual memory and proprioception to guide their behavior (see e.g. *Bardunias and Su, 2009a*), but these individual abilities are considered not sufficient to explain nest construction more generally. Instead, it is believed that building activity is largely guided by signals and cues embedded directly in the structure of the nest itself, through a regulation principle identified for the first time by Grassé, who named it *stigmergy* (*Grassé, 1959*; *Camazine et al., 2001*). In stigmergy-mediated nest-building, the probability for an individual insect to pick or to drop a pellet at a particular location is modulated by stimuli encountered at that location, such as the geometry of a growing pillar, or the presence of a chemical signal released by the queen or by other workers.

However, several years since Grassé's early observations, there is still not a consensus on the exact nature of the stigmergic stimuli involved in regulating termite construction. Pheromones might be implicated in this regulation. *Bruinsma, 1979* found evidence for the role of a building pheromone released by the queen in the construction of the royal chamber of the termite *Macrotermes subhyalinus*. Computer simulation studies, aimed at reproducing the building behavior of termites and ants, also assume the existence of a "cement pheromone" added to the building material (*Khuong et al., 2011*; *Khuong et al., 2016*; *Heyde et al., 2021*). In these simulation studies, the main and essential role of a cement pheromone is to allow initial pellet depositions to continue growing by differentiating them from regions of pellet collection, through differential pheromone marking. Experimental evidence in support for such cement pheromone in termite construction is weak: while individual workers can recognize freshly deposited nest material, they could simply be attracted to an unspecific colony odor while exhibiting the same behavioral patterns that they would exhibit also in the absence of chemical marking (*Petersen et al., 2015*). In other words, it is not clear if cement pheromones are required to drive termite building activities, or unspecific chemical cues would be sufficient, and it is also unclear if chemical stimuli modulate the building behavior of termites directly, or only indirectly, by affecting their density of presence.

Recent experimental studies by various authors have indicated that morphological and environmental features associated with some nest structures are strong stimuli that could by themselves guide termite construction activity. These include elevation (*Fouquet et al., 2014*), humidity gradients (*Soar et al., 2019*), and surface curvature (*Calovi et al., 2019*). The strong attractiveness of digging sites for termite aggregation means that in all these studies digging and deposition actions mostly come in pairs, which prevents us from identifying the genuine cues for pellet collection and deposition (*Bardunias and Su, 2009b*; *Bardunias and Su, 2010*; *Fouquet et al., 2014*; *Green et al., 2017*). For example, in *Calovi et al., 2019* termites are shown to preferentially aggregate in concave regions of a surface and they would simply rearrange nest material (both digging and building) at those locations. Even if digging sites provide a template for pellet deposition (*Fouquet et al., 2014*; *Green et al., 2017*), and for this reason digging and construction often co-localize in space, it is clear that building and digging cannot completely overlap, or the two activities would simply cancel one the effect of the other: termites must be able to differentiate between the sites of these contrasting activities through digging- or building-specific cues.

Some of the published computer simulation models of termite nest-building do not require a specific construction pheromone and assume instead that termites respond to cues naturally embedded in the nest structure itself. For example, the model proposed in *Ocko et al., 2019* indicates that a generic 'colony odor' undergoing advection and diffusion within the nest could provide a sufficient cue for determining the overall mound shape, so leaving a possible role of a construction pheromone only for the structuring of small scale nest features such as pillars and walls. *Facchini et al., 2020* further proposed a model in which also small scale nest features can be produced in the absence of a construction pheromone, by assuming that termites respond to the local curvature of these emerging nest features. While these models reproduce a number of structures observed in real termite nests, the building rules implemented in the models are not empirically validated from direct observations of the building behavior of termite workers. As it stands, there is no conclusive evidence that the rules implemented in these models reflect the actual nest-building strategies of termites.

Here, we aim to test whether geometric and physical cues embedded in the nest material are sufficient to explain termite construction. Specifically we want to disentangle how elevation, surface curvature, and substrate evaporation affect pellet deposition and collection. We do this by combining

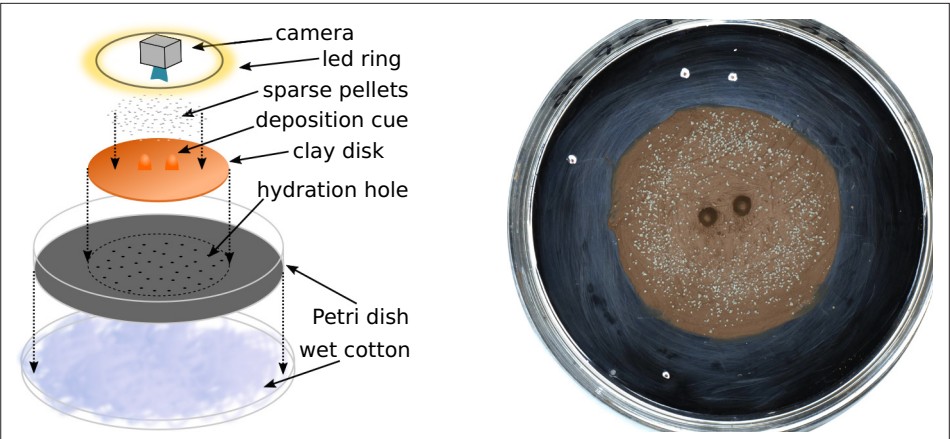

**Figure 1.** Experimental setup.
Sketch of the experimental setup (left) and snapshot of one experiment (E66) before termites were added to the setup (right). The white marks on the picture give the scale of the setup, with the distance between successive marks being 1, 3, and 5 cm.

The online version of this article includes the following figure supplement(s) for figure 1:

**Figure supplement 1.** Close-up on pillar structures built by our captive colony of *Coptotermes gestroi* within the plastic barrel that hosts the full colony.

**Figure supplement 2.** Temperature (blue) and humidity (red) within the Petri dish as a function of time.

three different approaches. (i) We perform building experiments in which populations of termites are confronted with pre-existing building cues such as pillars, walls, and pre-made pellets of building material unmarked with pheromones. Using video-tracking, we monitor the presence of individual termites and we implement high-throughput video-analysis to detect the time and location of individual pellet collection and deposition events. These experiments allow us to test the specific role played by each cue on stimulating pellet collection, pellet deposition, or termite aggregation. (ii) By running a computational model of nest building (*Facchini et al., 2020*) directly on the same structures that we provide to termites (obtained from 3D scans of our experimental setups), we can test exactly what building patterns we should expect under the simple assumption that termite depositions are driven by the local curvature of nest surfaces as the only construction cue. (iii) Finally, we develop a 'chemical garden' experiment, on identical setups to those offered to termites, that allow us to visualize the sites of stronger water evaporation on the surface of the built structure. Overall, our approach allows us to demonstrate, both analytically and experimentally, the relation between deposition probability, surface curvature, and evaporation.

## Results

Below, we report the observations of de novo building experiments performed with small experimental groups of *Coptotermes gestroi* termites confronted with a thin disk of humid clay covered with pre-made pellets unmarked with pheromones and decorated with pre-prepared clay features. In the first series of experiments the pre-prepared features were two pillars at the center of the clay disk as shown in *Figure 1*.

Pellet collection activity was distributed homogeneously all over the clay disk that we provided at the center of the experimental arena. Conversely, deposition activity was concentrated at the tips of pre-existing pillars, and along the edges of the clay disk itself. *Figure 2A* reports the heatmap of cumulative depositions $P(D)$ and collections $P(C)$ for one experiment (E66) with two pillars as topographic cues. A snapshot of the same experiment is reported in *Figure 3A* while a panoramic of all the experiments of this type is shown in *Figure 3—figure supplement 1*. In *Figure 2B*, we also report the same results for five experiments for which our analyses were most reliable because of the absence of spontaneous digging. Across all experiments, collections were widely distributed across the clay disk (i.e. where initial pellets are) while depositions were peaked at radii $R \sim 0.4\,\text{cm}$ and $R \sim 2.5\,\text{cm}$ which

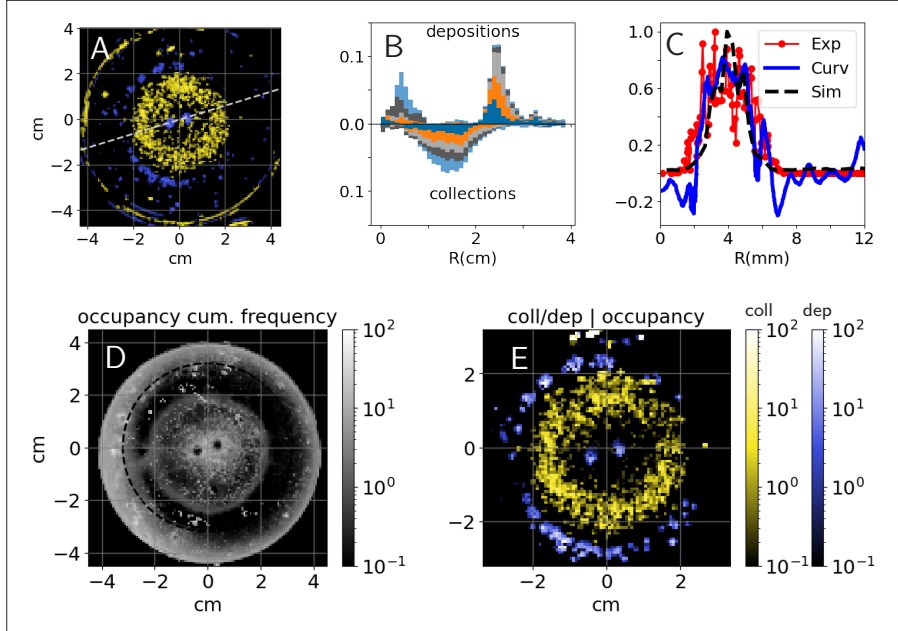

**Figure 2.** Experimental results - heatmaps. Top: (**A**) cumulative heatmaps of deposition ($P(D)$; blue) and collection activity ($P(C)$; yellow) normalized by their respective mean values for one experiment (E66), colorbars are the same as in panel (E); (**B**) cumulative depositions (top) and collections (bottom) per unit area as a function of the Petri dish radius for experiments E58, E63, E65, E66, and E76, all histograms have been normalized and sum up to 1; (**C**) comparison among experimental depositions (in red), surface curvature (in blue) shown in **Figure 3C**, and depositions predicted by simulations (black) shown in **Figure 3C**, all the quantities are computed along the radial cut shown in panel (A), depositions are normalized by their maximum value and curvature is in $\text{mm}^{-1}$; (**D**) cumulative occupancy heatmap normalized by its mean value for E66; (**E**) depositions ($P(D|O)$; blue) and collections ($P(C|O)$; yellow) conditional to cumulative normalized occupancy for E66.

correspond to the top of pillars and to the edges of the experimental arena. Thus, termites do not show a preference for where they collect pellets while they target specific regions when depositing, which suggests that those regions must express a strong stimulus for deposition.

To validate this hypothesis, we analyzed how building activity is related to the termite occupancy in the experimental setup. In **Figure 2D**, we report the normalized cumulative occupancy of termites $P(O)$ in the experimental setup. Occupancy is high close to the pillars and to the Petri dish walls, has intermediate values within the clay disk, and drops at the top of the pillars and right outside of the clay disk (i.e. precisely where deposits are recorded). To estimate how position and building activity are related, we report the conditional probabilities of depositing $P(D|O)$ and collecting $P(C|O)$ given termite occupancy. They are defined as the ratio between $P(D)$ to $P(O)$ and $P(C)$ to $P(O)$ as reported in **Figure 2E**, and explained in Materials and methods. The probability $P(D|O)$ reaches values 10 times larger than $P(C|O)$ which confirms that our topographic cues and the clay disk edges specifically drive early building activity.

Focusing on topographic cues, we observe that pillar tips are the most curved part of the topography but also the most elevated one. In order to disentangle the respective roles of curvature and elevation in guiding pellet deposition, we considered a different setup where a thin wall replaced the two pillars in the center of the arena as shown in **Figure 3B** (a panoramic of all the experiments of this type is shown in **Figure 3—figure supplement 2**). This way, the top edge of the wall is still a region of both high elevation and high surface curvature but elevation is constant everywhere while curvature has local maxima at the tips. We report that the top edge attracted many deposits, but pellet deposition focused at the wall tips pointing to curvature, rather than elevation, as the dominant cue in 7 out of 11 experiments (**Table 1**).

We wanted to further test to what extent the patterns that we observe are consistent with termites only responding to local substrate curvature, as opposed to responding also to other cues. To this end, we ran a model of nest construction that we have previously developed (**Facchini et al., 2020**) using

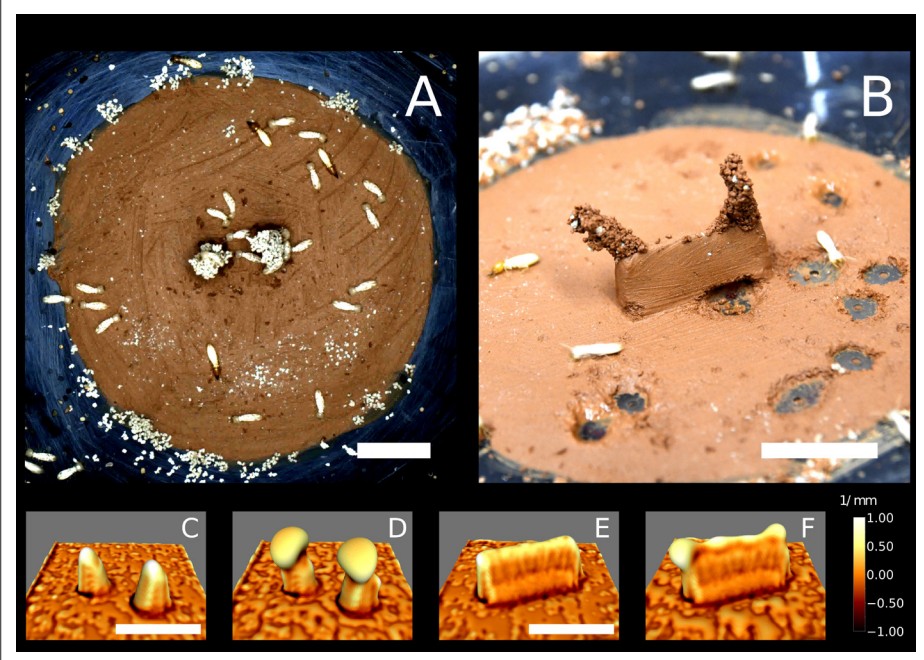

**Figure 3.** Laboratory experiments and simulations of the growth model.
Top row: snapshots of a building experiment with 'pillars' cue (E66) (**A**) and a building experiment with 'wall' cue (E78) (**B**). Bottom row: snapshots of 3D simulations initiated with copies of the experimental setup E66 (**C, D**) and E78 (**E, F**) in which nest growth is entirely determined by the local surface curvature (based on our previously described model *Facchini et al., 2020*). Snapshots C and E refer to t=0, D and F refer to t=9 (dimensionless). The color map corresponds to the value of the mean curvature at the interface air-nest. White indicates convex regions and black indicates concave regions. The scale bars correspond to 1 cm.

The online version of this article includes the following figure supplement(s) for figure 3:

**Figure supplement 1.** Snapshots of experiments with pillar cues.

**Figure supplement 2.** Snapshots of experiments with wall cues.

**Figure supplement 3.** Snapshots of experiments with no cues.

3D scans of the experimental arena – before the introduction of termites – as a starting template for the simulations (see Materials and methods for details). The simulation model implements one single construction rule which is a building response to local surface curvature and as such informs us about the possible building outcome that we could expect under the simplified assumption that construction is driven by surface curvature only, in the absence of any other cues. We should note that our model is phenomenological, and having one single parameter –that broadly defines the scale at which curvature is sensed– is not intended to predict quantitative details such as the speed of the building process, or fine details of the shape of pillars and other structures. These simulations yielded the results shown in *Figure 3C–F*. Experiments and simulations show a fair agreement as pellet depositions and initial growth concentrate in the same regions which are those where the surface is the most convex, as depicted in white in *Figure 3C–F*. To make our comparison more readable, in *Figure 2C* we report a radial cut of the deposition heatmap (red), the surface curvature (blue) and the amount of depositions predicted by the simulations (black; see Materials and methods for the technical details of this comparison). The three curves show a good agreement and they all are peaked close to $R = 4\,\text{mm}$ which corresponds to the pillar tips. The qualitative agreement between experimental results and curvature-based simulations supports the idea that, at least on a first approximation, surface curvature alone is a sufficient cue that could guide termite depositions.

The edges of the clay disk were not included in the simulations because we could not characterize them properly with our scanning device (see Materials and methods). However, in additional experiments with no topographic cues (see *Figure 3—figure supplement 3*) most depositions happened precisely at the edges of the clay disk. It is possible that the very small edge of the clay disk provided

**Table 1.** Summary table of experiments.

Color labels in the leftmost column denote batteries of identical simultaneous experiments. Orange highlighting denotes wall experiments where deposits focused at the lateral tips of the top wall edge. Experiment IDs labeled with a * refer to cases where: the clay disk dried prematurely (57,74,75); hydrating holes were drilled on one half of the disk (76); a larger number of workers (n=150) was introduced in the arena.

| Exp ID | Cue type | Cues deposits (any time) | Edge eposits | Else where deposits | Spontaneous holes |
|--------|----------|--------------------------|--------------|---------------------|-------------------|
| 48 | Pillars | Yes | No | Minor | 1 |
| 52 | Pillars | Yes | No | No | No |
| 53 | Pillars | Yes | Yes | Some | No |
| 57* | Pillars | Minor | No | Some | Yes |
| 58 | Pillars | Yes | Yes | Minor | No |
| 59 | Pillars | Some | No | minor | Yes |
| 63 | Pillars | Yes | Yes | No | No |
| 64 | Pillars | Yes | Yes | No | 1 |
| 65 | Pillars | Yes | Yes | No | 1 |
| 66 | Pillars | Yes | Yes | No | No |
| 67 | Pillars | Yes | Yes | Yes | Yes |
| 72 | Pillars | Yes | Yes | No | No |
| 73 | Pillars | Yes | Yes | Some | Yes |
| 74* | Pillars | Yes | Yes | Yes | Yes |
| 75* | Pillars | Yes | No | Yes | Yes |
| 76* | Pillars | Yes | No | No | No |
| 49* | Wall | Yes | No | No | n/a |
| 71 | Wall | Some | Yes | Minor | No |
| 77 | Wall | Yes | Yes | Some | Yes |
| 78 | Wall | Yes | Yes | No | No |
| 79 | Wall | Yes | Yes | Minor | No |
| 81 | Wall | Yes | Yes | Some | Yes |
| 90 | Wall | No | No | Yes | Yes |
| 91 | Wall | Yes | Yes | No | No |
| 92 | Wall | Yes | Yes | Minor | Yes |
| 93 | Wall | Yes | Yes | No | 1? |
| 94 | Wall | Yes | Yes | Minor | Yes |
| 83 | None | Spont. pillars | Yes | Minor | Non |
| 84 | None | n/a | Yes | Minor | Yes |
| 85 | None | n/A | Yes | No | No |
| 86 | None | Spont. pillars | Yes | No | No |
| 87 | None | n/A | Yes | Some | Yes |

a sufficient stimulus, in terms of local curvature, to elicit pellet depositions. However, the curvature cue was likely very weak at those locations as edges were smoothed out to gently match the surface of the Petri dish. Thus, we expect this region to bear a cue other than curvature (or elevation) which is also attractive for pellet depositions.

Trying to identify this additional building cue, we recall that the clay disk is maintained constantly humid. The edges of the disk arena mark then the limit between a humid region and the surrounding dry periphery. Also, the clay tone remained unchanged during experiments which suggests that moisture is constantly evaporating from the clay disk while being replenished in water from below, and that the overall process is stationary. To confirm this hypothesis, we measured the value of humidity and temperature both in the central and peripheral regions. We observed a net increase in humidity and a net decrease in temperature coming from outside to inside the clay disk which is the footprint of evaporation (*Figure 1—figure supplement 2*). Inside and outside the clay disk, both quantities remained relatively stable indicating that the system is roughly in a steady state. Evaporation is a complex process, but close enough to the evaporating substrate, humidity transport happens by diffusion (*Langmuir., 1918*; *Hisatake et al., 1993*) and it is hence fully determined by the humidity gradient. In agreement with previous studies (*Soar et al., 2019*), we can show with scaling arguments that our termites live in such a *viscous boundary layer* (see Appendix 3). We can hence focus our attention on this specific region without loss of generality. For example, the humidity transition at the edges of the clay disk implies that the humidity gradient must be pronounced there and evaporation with it. In the diffusive regime, the evaporation flux is directly proportional to the surface curvature of the

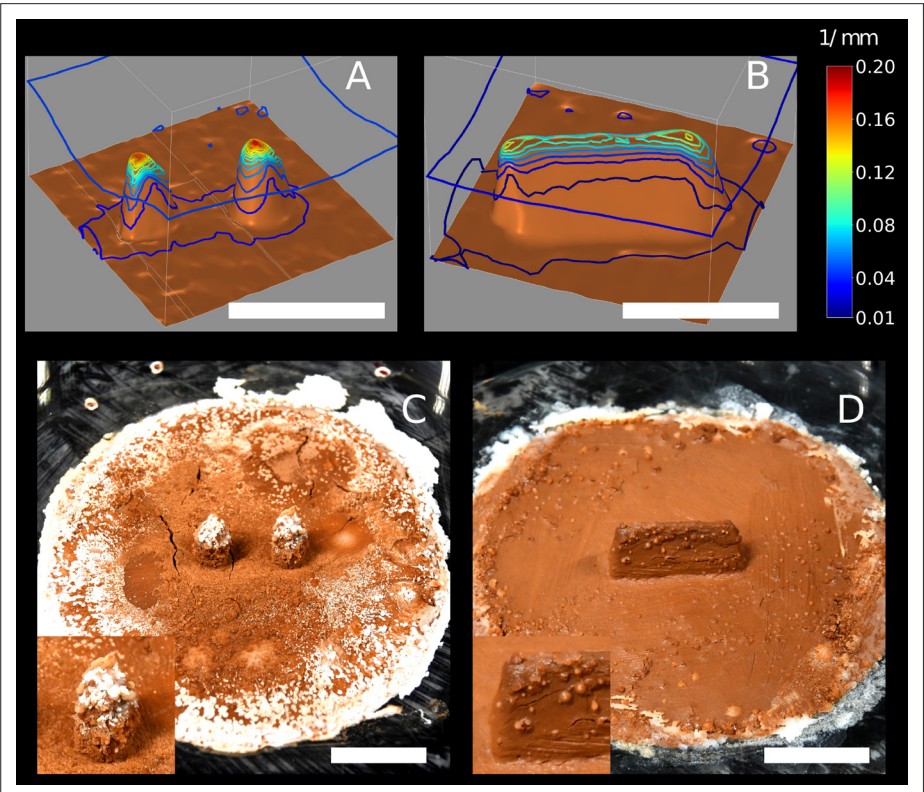

**Figure 4.** Diffusive simulations and chemical garden experiments. Top row: contour of the humidity gradient $\nabla h$ obtained solving the Laplace equation $\Delta h = 0$ in a cubic domain with a humid bottom boundary $h = 100\%$ (in brown) which is mapped from 3D scans of the experimental setup in E66 (**A**) and E78 (**B**). At the top boundary $h$ is fixed to $h = 70\%$ which was the average value of humidity in our experimental room. Note that $h$ is the relative humidity, thus the magnitude of the humidity gradient $|\nabla h|$ is measured in $\mathrm{mm}^{-1}$, i.e $|\nabla h| = 0.1\,\mathrm{mm}^{-1}$ means a humidity variation of 10% over $1\,\mathrm{mm}$. Each contour corresponds to a variation of $0.015\,\mathrm{mm}^{-1}$. Pillar tips are associated with a strong humidity gradient; the top of the wall is also associated with a strong humidity gradient, although not as strong as at the pillar tips. Also note that humidity gradient at the top corners of the wall is 30% stronger than on the rest of the top edge. Bottom row: snapshots of chemical garden experiments initiated with 'pillars' cue (**C**), and with 'wall' cue (**D**). All the scale bars correspond to $1\,\mathrm{cm}$.

evaporating substrate (see Appendix 2 for a mathematical proof). As a demonstration for our topographic cues, we have computed the steady state solution for the humidity field $h$ in a cubic volume bounded by pillars and wall experimental templates at the bottom. In the diffusive regime, this corresponds to solving the Laplace equation $\Delta h = 0$ while imposing a relative humidity $h = 100\%$ at the bottom boundary, and $h = 70\%$ at the top boundary (see Materials and methods for details) which was the average value of $h$ in our experimental room. In *Figure 4A and B*, we have reported the contour plot of the magnitude of the humidity gradient $|\nabla h|$ for this stationary solution. We can observe that the humidity gradient is maximum at the tip of pillars and at the lateral tips of the wall top edge, that are the most curved parts in the two different cues (see *Figure 3C and E* for a direct comparison). As such, curvature and evaporation are two completely interchangeable stimuli everywhere except at the edge of the clay disk, where the transition between clay and perspex material corresponds to a strong humidity gradient in spite of weak surface curvature. We then propose that evaporation flux can explain by itself the deposition patterns observed in our experiments.

To support this hypothesis, we designed a chemical garden experiment that allows us to visualize the evaporation field in our setup. We prepared identical experimental setups as those used with termites, but this time we did not put any termites in the experimental arena. Instead, we replaced the deionized water that was used to humidify the clay in the termite experiments with a saturated saline solution of water and $NaHCO_3$. In this configuration, water evaporation is accompanied by the deposition of salt, which allows to build a chart of evaporation flux. Typical results are shown in *Figure 4C and D*. Salt deposits appear in the form of white traces or bumpy deformations of the clay surface. Remarkably the distribution of the salt deposits matches very closely the regions of highest building activity by termites, both being more pronounced at the edge of the clay disk and at the top of topographic cues (pillar and walls). This result indicates that termite deposition probability covaries with the evaporation flux, which is consistent with our hypothesis of evaporation as the strongest cue for deposition. One may notice that salt traces are less pronounced on the wall (*Figure 4D*) than on the pillars (*Figure 4C*). This is consistent with the amplitude of the humidity gradient in a stationary diffusive regime as shown in *Figure 4B and A* respectively: the maximum amplitude of the gradient is weaker in the wall case. Also, one observes that while having its maxima at the lateral tips ($|\nabla h| \sim 0.13\,\mathrm{mm}^{-1}$), the humidity gradient is strong all over the top edge of the wall ($|\nabla h| \sim 0.10\,\mathrm{mm}^{-1}$). Possibly, this could explain why in the case of wall cues there is a comparatively much larger variability in the patterns of termite construction (see *Figure 3—figure supplement 2*). When termites are confronted with the 'wall' cue they are likely to start pellet depositions at the wall tips, but initial pellet depositions started at other locations on the top edge do also occur. Whatever the initial choice, depositions are then preserved and reinforced by positive feed-back mechanisms which can lead to tightly selected deposition patterns also in the case of the 'wall' cue.

Note that, in the picture of depositions being attracted by evaporation flux, depositions observed at the edges of the clay disk, also agree with our previous growth model driven only by curvature (*Facchini et al., 2020*). In fact, the edge of the clay disk is almost flat (weak convexity) for a termite walking across, but it is also a thin cusp (high convexity) of humid material which is strongly evaporating, similarly to what happens at the edge of a liquid drop and causes the formation of well known coffee stains (*Deegan et al., 1997*). This apparent contrast is explained in the sketch of *Appendix 2—figure 1*.

## Discussion

Several experimental studies have tried to identify the cues that mediate termite construction, alternately indicating elevation (*Fouquet et al., 2014*), digging activity (*Green et al., 2017*), humidity transitions (*Soar et al., 2019*; *Bardunias et al., 2020*), or surface curvature (*Calovi et al., 2019*) as the relevant stimuli to drive pellet depositions. However, the fact that termites often concentrate their building activity in the immediate proximity of digging sites (*Fouquet et al., 2014*; *Green et al., 2017*; *Bardunias and Su, 2009b*; *Bardunias and Su, 2010*) did not allow identifying which of these stimuli were specific digging and building cues, or simply generic cues for termite activity and aggregation. The cues themselves identified by different studies were different, leaving it unclear which, if any, were the relevant ones sensed by termites.

Here, by providing loose and unmarked pellets, we were able to prompt building activity without digging and to quantify collections and depositions as separate actions. We observed that all pellets

were progressively displaced and that collections happened in a relatively random fashion. On the contrary, depositions concentrated at specific parts of the experimental arena which are the tips of the topographic cues and the edges of the clay disk. The conditional probability of deposition given termite occupancy was high there, indicating that those regions precisely drive termite depositions rather than generically attracting termite aggregation. The alternative use of pillars and walls as topographic cues allowed us to disentangle the role of elevation and curvature and pointed toward curvature as the most attractive stimulus for deposition. By simulating the building process with a model in which construction activity is driven by curvature only (*Facchini et al., 2020*), we obtained a good match with experimental results, indicating that curvature alone is sufficient to explain pellet depositions on topographic cues (pillars and walls).

Surface curvature is a powerful morphogenetic organizer for 3D structure formation as it can drive the formation of pillars, walls and convoluted surfaces (*Facchini et al., 2020*), all features that are observed in the nests of various termite species. Here, we were able to demonstrate a close coupling between surface curvature and the flux of evaporation from a surface, so providing a link to a possible stimulus sensed directly by termites. This also allows us to reconcile previous discordant results pointing alternately to curvature (*Calovi et al., 2019*; *Facchini et al., 2020*) or to humidity (*Soar et al., 2019*; *Bardunias et al., 2020*) as the relevant stimuli. The idea itself of a relation between curvature and evaporation is not new, as already a century ago, *Langmuir., 1918* showed that close enough to the surface of a water droplet, evaporation scales as the inverse of the radius (i.e. as the mean curvature) of the droplet see (Appendix 2 for a derivation). Our system is more complex than isolated spheres but our calculations in Appendix 3 show that a relation between evaporation and curvature still holds at the termite scale.

As a further, direct, confirmation of our hypothesis, our chemical garden experiments clearly show that the correspondence between surface curvature and evaporation flux is relevant in our experimental setup.

It is well known that termites are particularly sensitive to the humidity of their environment, because their small size and soft cuticle put them in constant danger of desiccation. *Coptotermes* termites in particular are wetwood termites that can only survive in high-humidity environments such as moist wood or soil. For example, in laboratory experiments *Zukowski and Su, 2017*, *C. formosanus* termites died within a few days when maintained at 72.9% or less relative humidity, but survived well when humidity was 98%. It is hence not surprising that they can sense and respond to humidity with their behavior, for example (*Arab and Costa-Leonardo, 2005*) tested the digging behavior of *Coptotermes gestroi* in wet sand with different levels of moisture and showed that tunnel length and the number of secondary branches changed when soil moisture increased from 5% to 15%. *Coptotermes gestroi* were also able to discriminate between chambers with different relative humidity and, after 12 hr, almost all termites were in the highest humidity chamber (98%), leaving the other chambers with 75% or less relative humidity empty (*Gautam and Henderson, 2011*). These results (which are similar also to other results testing termite response to chambers with different soil moisture) indicate that –given a sufficient amount of time– termites can detect a difference of humidity from 75% to 98% over a spatial scale of centimeters. Recent field and laboratory experiments have shown that humidity can affect termite behavior also in the context of nest building (*Carey et al., 2019*), by triggering nest expansion events (*Bardunias et al., 2020*; *Carey et al., 2021*). Even more interestingly, *Soar et al., 2019* showed that moisture flux affects termite building activity (both in terms of pellet collection and deposition). Our experiments confirm this trend and suggest that moisture variations not only prompt or inhibit termite building activity, but constitute a local blueprint for construction.

One may question if the humidity gradient in our experimental setup was strong enough for termites to sense it. Below, we use simple arguments to show that this condition was very likely fulfilled. Indeed, one can estimate that the value of the humidity gradient far from the disk edges and the topographic cues was $|\nabla h|_0 = \delta h / \delta \sim 0.15 \, \text{mm}^{-1}$, where $\delta = 2 \, \text{mm}$ is the thickness of the diffusive boundary layer (see Appendix 3), and $\delta h = 30\%$ is the humidity variation between the clay disk surface and the experimental room.

Then, using the qualitative results of chemical gardens shown in *Figure 4C and D*, one can conclude that $|\nabla h|_0$ represents an estimation of the lower boundary of the humidity gradient experienced by termites in our experimental setup. Using diffusive simulations reported in *Figure 4A and B* (see also

Materials and methods), one can then quantify the relative importance of the humidity gradient at the tips of the topographic cues as 10 times larger, which gives an upper boundary $|\nabla h|_{max} \sim 1\,\text{mm}^{-1}$.

As our termites are millimetre size, the estimated lower and upper boundary correspond to humidity variation of 10 to 100% across a distance not larger than their body length. We then expect that such variations must be sensed by termites, as they are larger than those that *Coptotermes gestroi* termites were shown to be able to discriminate over much larger spatial scales in the experiments reviewed above.

Our experiments do not support a role for a putative cement pheromone, added by termites to the building material, which would stimulate pellet depositions. In fact, construction occurred reliably on our provided building cues, even if they only comprised fresh clay and sterilized pellets with no pheromone markings. Our simulations further indicate that qualitatively similar construction results can be obtained without assuming a role for construction pheromone. We can hence exclude the influence of a cement pheromone, at least during the early choice of the deposition sites, in agreement with recent experiments by other authors (*Fouquet et al., 2014*; *Petersen et al., 2015*; *Green et al., 2017*). We should point to the fact, however, that in our experiments the building substrate was constantly moist throughout the entire duration of the experiments. It is possible that in some occurrences of nest building behavior, including in termites' natural environment, moisture may not constantly replenish the porous wall of the growing structure. We suggest that under these conditions, the evaporation flux is maintained by the humidity that is naturally embedded in recently dropped pellets, which makes the construction process self-sustainable and is consistent with the hypothesis of a viscous boundary layer extending with termite activity (*Soar et al., 2019*). In practice, it would be very hard to distinguish between such a scenario and one which involves a putative cement pheromone added directly to manipulated pellets by termites. More generally, while we do not rule out a possible role of pheromones in termite building behavior (mediating for instance termite aggregation), we have shown that pheromones are not necessary to explain the early deposition patterns that we see in our experiments.

In this study, we have focused on understanding how termites respond to well-controlled predefined stimuli. However, collective nest construction is a dynamic process and the deposition of new pellets by termites constantly changes the shape and the porosity of the evaporating substrate, potentially affecting nest growth through positive or negative feedback. Recent studies have shown that termites can control the size of the pellets used for nest construction, and indirectly also the porosity of nest walls (*Zachariah et al., 2017*). In turn, substrate porosity is known to play an important role for ventilation and drainage of the nest (*Singh et al., 2019*) and the moisture content of pellets can also affect the mechanical properties of the mound itself (*Zachariah et al., 2020*). In relation to our own experiments, however, our scaling analyses (Appendix 3) indicate that our conclusions are relatively robust to changes in substrate porosity and moisture content. For example, porosity only controls the time scale of water uptake from the reservoir by capillary rise, which must be small enough to keep the clay disk hydrated, and this assumption remains valid up to mm-size pores in the new construction. Similarly, for local curvature, the addition of new pellets to regions of high convexity is likely to make the surface less smooth than the initial topography, and such additional 'roughness' can only increase the effect of focusing evaporation at those locations.

Previous work by *Calovi et al., 2019* had pointed to an effect of surface curvature on termite construction behavior. While our two studies emphasize the same point, we should note that our results and the results reported in *Calovi et al., 2019* are not entirely consistent, because in our experiments, pellet depositions are attracted by convex features, while in *Calovi et al., 2019* termite activity was concentrated at regions of maximum concavity. As this previous study did not distinguish digging from deposition activity, we believe that their measure is a correlation between concavity and digging activity, not building. The fact that concave regions should attract digging activity is predicted by our model (see Materials and methods) and was visible also in our experiments where concavity (*Figure 3D*) attracted digging at the base of wall-like cues (*Figure 3—figure supplement 2*). Note that such behavior can be interpreted as termites digging along the humidity gradient, that is toward the most humid region. Accordingly, in many preliminary experiments we observed that, in the absence of loose pellets, spontaneous digging usually started right above the hydration holes of our setup (*Figure 3B*) as a further confirmation of the termite ability to sense and respond to humidity gradients.

In our study, we have outlined a general mechanism capable of organizing termite building activity: termites would focus pellet depositions at regions of strong evaporation flux. In turn, evaporation flux co-varies with surface curvature, which implies that the building rule is embedded in the shape itself of the nest internal structure.

One may wonder to what extent the simple building rule that we identify here generalizes to explain the nest-building behavior of larger termite colonies in the field, and whether the same building rules are shared across different termite species. The nests built by termites of different genera or species show a large diversity of forms (see e.g. *Grassé, 1984*), which indicates that the nest-building process should also be different. Arguably, the nest building behavior of termites, shaped by millions of years of evolution, must rely on more complex 'building rules' than the simple ones highlighted here. Nonetheless, it is interesting to notice that the nests built by all species rely on a small number of architectural elements such as pillars and branching surfaces. We can imagine that, perhaps, simple variation of the basic building pattern described here, coupled with variation of the substrate evaporation itself (e.g. under the effect of air currents, the properties of the building material, and heat produced by the colony itself) would still be sufficient to explain a large part of termite nest diversity. *Ocko et al., 2019* have already shown that a single mechanism can be responsible for determining the overall shape of nests made by various species: perhaps an equally simple general mechanism can account for the even larger variation of internal nest structure.

Beyond the field of termite architecture, many other biological structures, particularly at the microscopic scale, present convoluted structures that appear via tip growth and branching phenomena. Similar to termite nests, the morphogenesis of these structures may rely on the coupling between the local curvature of the growing substrate and the localization of cues that stimulate the growth. Generally, there is a growing consensus today on the fact that local curvature can affect cell migrations, cell patterning and tissue growth typically by direct mechanical sensing through the cytoskeleton (*Pieuchot et al., 2018*; *Callens et al., 2023*). However, analogous mechanisms to those described here can also explain tip growth and branching in mammal lungs, *Clément et al., 2012*, and reconnections in the gastrovascular networks of jellyfishes *Song et al., 2023*, the common feature being that curvature guidance acts indirectly, by focusing the gradient of a diffusing quantity that promotes the substrate growth.

## Materials and methods

In our experiments, we monitor the building behavior of small experimental groups of *Coptotermes gestroi* termites confronted with a thin layer of clay and pre-prepared clay features. We image experimental trials for their entire duration and we analyze termite activity with custom-made digital image processing routines which are described below. In parallel, we run two types of control experiments without termites to obtain a non-intrusive estimation of temperature, humidity and evaporation field in our experimental setup. These experiments are described at the end of this section. Finally, we perform numerical simulations of a growth model and a diffusive model to support our experiments. Their implementation is also described is this section.

### Termite species

In our experiments, we studied the building behavior of termites *Coptotermes gestroi*. *Coptotermes gestroi* is a soil-nesting wood-feeding termite originating from South-East Asia (*Li et al., 2010*). However, as a result of the increase of global human activity, it has spread in Asia and to other parts of the world including Africa, Europe, North America, Central America and South America (*Su et al., 2017*). As an exotic pest it is primarily found in populated urban environments, where it feeds on human-made structures (*Su et al., 2003*). In its natural habitat *C. gestroi* feed on dead trees and wooden debris on the soil surface and build their nest underground, although 'aerial' infestations have also been recorded in human made structures, where the nest have no contact with the ground. Whether the nest is aerial or subterranean, the internal nest structure is similar to the nests of other termites in the *Coptotermes* genus, and comprises a 'scaffold' of interconnected pillars (see *Figure 1— figure supplement 1*). This structure is the result of a building process, as evidenced by the fact that its material composition is different from the composition of the surrounding soil and comprises some

stercoral carton. Colonies of *C. gestroi* are estimated to be within the range of 100,000–4,000,000 individuals (*Ab Majid and Ahmad, 2015*).

### Captive colonies

For most experiments, termites were collected from the same master captive colony (c22) of *Coptotermes gestroi* hosted at the LEEC laboratory (Villetaneuse, France) in a tropical room with constant temperature (26 ± 2°C) and relative humidity (70 ± 10%). Only experiments 57, 90, 91, 92, 93 and 94 were run using termites from a second master colony (c21). Experiments were performed in the same tropical room in which the master colonies were housed to reduce the environmental stress on the experimental populations. Workers were attracted with humid towels and gently shoveled with a pencil on a plastic tray. Groups of 50 workers and 5 soldiers were then formed using insect forceps and added to an experimental setup. While the procedure might be potentially stressful to termites, mortality was negligible throughout all the experiments.

Soldiers were mostly inactive during all our experiments but were included to maintain the same proportion as in a real colony (1 soldier for every 10 workers). This choice was made to limit as much as possible the factors that might affect building behavior for not being in a natural situation. Finally, when running batteries of multiple experiments, distinct groups of 50 workers were formed by rounds of ten individuals taken individually from the plastic tray. This choice was made to avoid introducing any unwanted bias in the groups, coming from the fact that inactive or larger termites are usually picked first from the tray because they are easier to pick.

We ran 16 experiments with pillar cues, 11 with a wall and 6 with no cues, as summarized in *Table 1*. An example of the structures spontaneously built by these termites within the plastic barrel which hosts the captive colony is shown in *Figure 1—figure supplement 1*.

### Experimental setup

The experimental setup is sketched in *Figure 1* (left) and can be described as follows. A fixed quantity (2.8 g) of red humid clay paste is flattened to form a disk (5 cm) and placed in the center of a Petri dish (8.5 cm). A system of 40 small holes (0.8 mm) drilled in the bottom of the Petri dish keeps the clay paste hydrated sucking distilled water from a patch of wet cotton below the Petri dish. Two types of topographic cues molded in clay, can then be added at the center of the disk: 2 pillars 6 mm high and 8 mm apart or 1 wall 6 mm high and 12 mm long. The pillars are obtained pressing clay in a small eppendorf tube. The walls are obtained by smoothing a wedge of clay generated by rolling out a piece of clay in the dihedron between the table and the edge of a plastic ruler. Finally, in a circular band (1 cm large) halfway between the clay disk center and its edge, we add 0.12 g of sparse pellets of gray clay. To ensure the good size distribution, pellets are obtained from previous experiments and sterilized at 100 $^{\circ}C$ for 1 hr to remove any possible chemical marker. Before the start of each experiment, a surface scan of the setup was taken using a NextEngine 3D Scanner ULTRA HD.

### Recordings

A led lamp constantly lightened the setup from above. Top view pictures of the experimental setups were taken at regular intervals of 20 s during at least 24 hr using a reflex camera Nikon D7500. By subtracting the initial images and applying a median filter, we get rid of termites and the background as detailed below. Then, the color contrast between the clay disk and the pellets allowed identifying where pellets were collected (dark spots) and deposited (bright spots) and building the heatmap of both activities as a function of time. A subset of the experiments was recorded continuously using a 12 Mp usb-Camera (MER2-1220-32U3C) at 7fps.

### Collection and deposition heatmaps

We developed an image processing pipeline that allows tracking the displacement of pellets in the arena, that is quantifying the collection and deposition of material. First, we consider the time-lapse or video recording and extract a sequence of images at intervals of 80 s, which is a rough estimation of the time of a search-collection-deposition sequence. Images are then converted to greyscale and the image of the initial setup (without termites) is subtracted from all of them. At the next step, we replace each image in the sequence by the median across 10 consecutive images, that is each pixel takes the median value across 10 consecutive images. This operation is meant to take rid of termites that are

constantly moving while pellets are usually moved only once, from the collection site to the deposition site. Notice that pellets are light grey while the arena is brown, so after subtracting the initial image, any collected pellet leaves a dark (low luminosity) trace in the sequence, while any dropped pellet leaves a bright (high luminosity) trace in the image sequence. Subsequently, we average the sequence across intervals of 10 images, thus integrating termite activity in windows of 10*80=800 s.

A demo of a typical image sequence at this stage is available at this link. To quantify the amount of collected pellets, we then binarise each image at low threshold and isolate collected pellets as black regions so that they can be labeled with a connected components algorithm. We exclude components whose area is below 10 px (i.e. half the area covered by the smallest pellets) or above 400 px. The lower and upper thresholds was meant to get rid of noise and clusters of inactive termites, respectively. Finally, we convert again binary images to greyscale images, and re-scale each image so that one pixel is 0.75 mm large which is the size of an average pellet. This last operation allows reducing significantly the size of our heatmap without loosing information. We have then obtained a heatmap of collection activity as a function of time. An equivalent result is obtained for the spatial distribution of depositions, simply inverting the pixel values of initial images. As a matter of fact, most of the pellets are displaced only once, thus looking at the last heatmap in the temporal sequence, one has a good estimation of the cumulative distribution of collections and depositions across the experiment: we refer to these cumulative distributions as $P(C)$ and $P(D)$ throughout the manuscript.

## Termite occupancy

Termites motion is analyzed using the fast multi-animals tracking tool TRex (**Walter and Couzin, 2021**). The ultimate aim of TRex algorithm is to identify the trajectories of each individual and record their position and velocity as a function of time. Unfortunately, TRex did not manage to preserve individual identities from our videos: the identity of single trajectories was rapidly lost and we could not rely on them to study individual termite behavior. While the identity of individual termites was not preserved, we verified that the trajectories tracked by the software truly reflect real termite locations, and as such they can be used to obtain information about the overall termite occupancy within the Petri dish. Once the trajectories are extracted by TRex, we proceed in the same way as with pellets traces, that is, we construct a heatmap of positions on the same regular grid (0.75 mm grid step) and during 800 s intervals.

## Conditional probabilities

To assess the randomness of collection and deposition events, we wanted to estimate the probability of these two events at a given position conditional on the probability of termite occupancy at the same position. Let us denote with $P(C|O)$ the conditional probability of collection given occupancy, then by definition:

$$P(C|O) = \frac{P(C \cap O)}{P(O)}.$$

However, a collection episode necessarily happens where a termite is, which means that collection events $C$ are a subset of occupancy events $O$, which implies:

$$P(C|O) = \frac{P(C)}{P(O)},$$

that is $P(C|O)$ is simply the ratio between collection and occupancy frequency. The same reasoning can be done for the conditional probability of deposition given occupancy $P(D|O) = P(D)/P(O)$ which is the ratio between deposition and occupancy frequency.

## Normalization of probability distribution

Note that in the plots of *Figure 2* the frequency heatmaps ($P(C)$, $P(D)$, and $P(O)$) do not sum up to 1. Instead, we preferred to normalize the probabilities by their average value, which allows us to easily recognize high-frequency areas where $P > 1$ and low-frequency areas where $P < 1$. Coherently with this choice, conditional probabilities ($P(C|O)$ and $P(D|O)$) are not normalized to 1 either.

### 3D surface scans

Before the start of each experiment, a surface scan of the setup was taken using a 3D surface scanner (NextEngine 3D Scanner ULTRA HD). This allowed initializing the numerical simulations of the growth and the diffusive model with the same topographic cues as in the experiments, as described below. The scanner resolution was sufficient to characterize the curvature of topographic cues, but it was insufficient to detect the edges of the clay disk. In this region, an additional problem came from the fact that the Petri dish (bare) surface could not be scanned properly, likely because of the laser getting reflected two times at the top and bottom side of the plastic plate. This effect, combined with the small size of these features relative to the spatial resolution of the scanner, made it difficult to reliably measure surface curvature at the edges of the clay disk.

### Temperature and humidity measurements

Temperature and humidity were measured using a commercial temperature-humidity probe (DHT22) connected to a Raspberry Pi micro computer. In order to not interfere with termite behavior, our temperature and humidity measurements were performed on a control experiment, which was prepared using the same protocol as the other experiments but where no termites were added. The results are reported in *Figure 1—figure supplement 2*. The shaded area corresponds to a time interval where the probe was placed on the bottom of the Petri dish, outside the clay disk, while the rest of the time the probe was placed on the clay disk. We can observe a net decrease in temperature and an increase in humidity when moving from inside to outside the clay disk, while the overall values within the two regions do not show large variation across several hours. This is consistent with the fact that the clay disk undergoes a stationary evaporation process and the well know cooling of the substrate due to the latent heat of water.

### Growth model

Our growth model is the same described in *Facchini et al., 2020* which consists in one single non-linear phase field equation:

$$\frac{\partial f}{\partial t} = -f(1-f)d\nabla \cdot \boldsymbol{n}f - \Delta\nabla \cdot \boldsymbol{n}. \tag{1}$$

where the scalar $f$ takes values between 0 and 1 and identifies the presence of the nest wall ($f < 0.5$) or the empty space ($f > 0.5$), and $\boldsymbol{n} = \nabla f/|\nabla f|$ is the normal vector at the wall surface $f = 0.5$. One recognizes a growth term proportional to the surface curvature $-\nabla \cdot \boldsymbol{n}/2$ which translates our main hypothesis on construction behavior, and a curvature diffusion term $\Delta\nabla \cdot \boldsymbol{n}$ which mimics the smoothing behavior shown by termites (see *Facchini et al., 2020* citations inside) and the fact that there is a cutoff to the size of pellets added by termites. Finally, the non linear prefactor $f(1-f)$ restricts the growth process to the wall surface, which is also coherent with termites behavior. Note that the simulations shown here are obtained approximating $\nabla \cdot \boldsymbol{n} \approx \Delta f$ as in *Facchini et al., 2020*. The parameter $d$ selects the typical length scale of the expressed pattern and the cutoff scale below which features are smoothed out. At present, our model is still phenomenological and we cannot provide an estimation of $d$ based on the experimental measurements of a limited set of physical quantities. Thus, we tune the parameter $d$ to select a typical length scale which matches that of our topographic cues, that is the thickness of our clay pillars and walls (3 mm), which in turn was chosen to be of the same order of the termite size.

Our simulation are initialized using 3D copies of the experimental setups at time zero that are obtained as it follows. First, we obtain a surface scan of the experimental setup in the form of a 3D mesh using a surface 3D scanner (see Materials and methods). Then, we interpolate the mesh on a 3D regular grid and assign the initial value of the scalar field $f$, setting $f = 1$ for the points that are below the clay surface and $f = 0$ for the points that are above the clay surface. Finally, a Gaussian filter is applied to unsharp the transition of $f$ at the surface. Similarly to our previous publication, we also assume that the voxels where $f > 0.85$ at $t = 0$ cannot change in the following. This translates the fact that structures built by termites are not observed to be rearranged after they have dried and that in our experiments termites are prompted to collect pellets instead of digging.

## Comparison between experiments and simulations of the growth model

*Figure 2C* reports the values of termite deposition activity from one experiment (red line, from experiment E66) alongside with the local surface curvature measured for the same experimental setup (blue line). The same figure panel also reports the deposition activity from a simulation of our 3D curvature-based growth model, initialized with the same topography as in experiment E66 (blue line). All values are plotted along a one-dimensional cross-section cutting across the pillars.

The data contributing to this figure panel originate from different sources: experimental pellet collection and deposition events are obtained from timelapse photographs, while curvature is obtained from 3D scans. Below we explain how we realign the data to take the 1D projection shown in the figure. For the experimental pellet depositions, we simply considered the value of the deposition heatmap shown in *Figure 2A* by selecting the cells adjacent to the black dashed line which is shown in the same figure. The value are then normalized by their maximum. For the curvature values, we took the values of curvature corresponding to the mesh elements of the 3D model shown in *Figure 3C* which fall close to the same line. In order to quantify pellet depositions from the 3D simulations, we proceed as follows: first, we identify all the voxels of added material by subtracting the original 3D shape from the final one. Then, we select all the points that fall close to the plane passing through the black dashed line shown in *Figure 2A*. For each of these voxels, we then identify a *corresponding* voxel on the original 3D shape which is the closest one. Finally, we compute the radial positions in the horizontal plane of all the *corresponding* voxels and we plot the histogram of these radial values, where the histogram is normalized by its maximum to obtain the 1D projection shown in *Figure 2C*.

## Numerical simulations of the humidity field

Below we discuss the implementation details and the approximation of the diffusive model used to predict the variations of the humidity gradient in our experimental setup as shown in *Figure 4A and B* of the manuscript. This model relies on the assumption (see Appendix 2 for a discussion) that humidity transport uniquely happens by diffusion and happens in a quasi-stationary way, which corresponds to solve the Laplace equation $\Delta h = 0$ for the humidity field $h$, which is done using the finite elements platform COMSOL Multiphysics. Our simulation domain was a cubic domain of side $18\,mm$ whose bottom face is replaced by a 3D copy of the experimental setup, in the case of pillars cue (*Figure 4A*) and wall cue (*Figure 4B*). The boundary conditions on the field $h$ were no-flux on the lateral boundaries and Dirichlet on the bottom and top plate where the relative humidity $h$ was fixed to 100% and 70%, to mimic the experimental value of $h$ respectively at the clay disk surface and in the air (far from the disk surface) of our experimental room. At a given temperature, relative humidity and absolute humidity are proportional, so assuming here that temperature is constant, we always refer to relative humidity and our simulations measure the humidity gradient at any position in units of $mm^{-1}$ (exactly like curvature).

## Saline solution experiments

We performed control experiments with no termites and hydrating the clay disk with a saline solution instead of distilled water to map the distribution of evaporation flux. Saline solution was prepared adding $8\,g$ of $NaHCO_3$ to 100 ml of tap water.

## Acknowledgements

We thank Paul Devienne at the LEEC laboratory for his help in taking care of termite colonies. We thank Baptiste Piqueret at LEEC laboratory for inspiring discussions. We thank Anand Douady for his inspiring experiments with salt and evaporation. This work was supported by a Royal Society Newton International Fellowship NIF\R1\180238 and by a Leverhulme Research Project Grant RPG-2021–196.

## Additional information

### Funding

| Funder | Grant reference number | Author |
|---|---|---|
| Royal Society | NIF\R1\180238 | Giulio Facchini |
| Leverhulme Trust | RPG-2021-196 | Andrea Perna |

The funders had no role in study design, data collection and interpretation, or the decision to submit the work for publication.

### Author contributions

Giulio Facchini, Conceptualization, Data curation, Formal analysis, Funding acquisition, Investigation, Methodology, Writing – original draft, Project administration, Writing – review and editing; Alann Rathery, Investigation, Methodology, Writing – review and editing; Stéphane Douady, Conceptualization, Formal analysis, Supervision, Writing – review and editing; David Sillam-Dussès, Conceptualization, Resources, Supervision, Methodology, Project administration, Writing – review and editing; Andrea Perna, Conceptualization, Resources, Formal analysis, Supervision, Funding acquisition, Investigation, Methodology, Writing – original draft, Project administration, Writing – review and editing

### Author ORCIDs

Giulio Facchini ⓘ http://orcid.org/0000-0003-4500-5748
David Sillam-Dussès ⓘ http://orcid.org/0000-0001-5027-8703
Andrea Perna ⓘ http://orcid.org/0000-0001-8985-3920

Joint Public Review: https://doi.org/10.7554/eLife.86843.4.sa1
Author response https://doi.org/10.7554/eLife.86843.4.sa2

## Additional files

### Supplementary files

• MDAR checklist

### Data availability

Semi-raw data and processing python routines are available on figshare with a detailed description. Raw high resolution timelapses and videos are not included under this link as they are too large, but these can be requested directly from the corresponding authors.

The following dataset was generated:

| Author(s) | Year | Dataset title | Dataset URL | Database and Identifier |
|---|---|---|---|---|
| Giulio F | 2024 | Post-processed timelapse and analysing algorithms | https://doi.org/10.6084/m9.figshare.20449716.v2 | figshare, 10.6084/m9.figshare.20449716.v2 |

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

## Appendix 1

### Curvature definition

In this section, we give a mathematical definition of *surface curvature* and we explain how its sign is determined. Given a three-dimensional object, its surface can be characterized locally using the reciprocal of the radius of two osculating circles $r_1$ and $r_2$ as sketched in *Appendix 1—figure 1*. One can then define the first principal curvature $k_1 = 1/r_1$ and second principal curvature $k_2 = 1/r_2$, or alternatively the mean curvature $H = (k_1 + k_2)/2$ and the Gaussian curvature $\Gamma = k_1 \cdot k_2$. Throughout the manuscript when using the term *curvature* or *surface curvature*, we always refer to the mean curvature $H$.

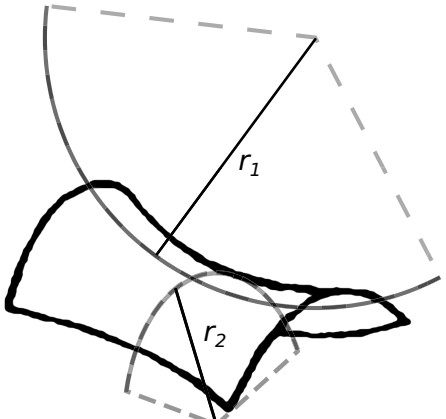

k1=1/r1:  1st principal curvature
k2=1/r2:  2nd principal curvature

Gaussian curvature : Γ = (k1*k2)
 mean curvature : H = (k1+k2)/2

**Appendix 1—figure 1.** Sketch of curvature definition on a saddle shaped surface.

Provided that we have established a convention for the interior and the exterior of the surface, i.e. once the normal vector $\hat{n}$ to the surface is drawn, principal curvatures come with a sign and so do $H$ and $\Gamma$. Indeed drawing the normal vector univocally fixes the value (and the sign) of $H$ in the form:

$$2H = -\nabla \cdot \hat{n}.$$

Traditionally, the normal vector is oriented towards the exterior, for example thinking about a solid, the normal vector usually points toward the outer space. When this convention is adopted (which is for instance what was done in reference *Calovi et al., 2019*), convex objects have negative curvature and concave objects have positive curvature. Conversely, here we use the same convention as in our phase field model (*Facchini et al., 2020*) which is the opposite one: convex surfaces have positive curvature and concave surfaces have negative curvature. While, of course, experimental results and theoretical predictions do not depend on which convention is adopted, the sign of the curvature can become important when comparing results across different studies. For example, *Calovi et al., 2019* suggested that termite building activity is enhanced in regions of high positive mean curvature, that -by their sign convention- are regions of highest concavity. In the present study, we also propose that positive mean curvature attract deposition of building material, but in our case the regions of positive curvature are the most convex. Thus, the two results do not agree. However, our model predicts that while convex regions attract deposition, concave regions attract digging, and the analysis techniques used by *Calovi et al., 2019* do not allow distinguishing between digging and deposition actions. We then propose that what was observed in *Calovi et al., 2019* is a correlation between curvature and digging activity which is in agreement with our model.

## Appendix 2

### Approximation of the diffusive zone

Note that the size of the simulation box in the simulations of our diffusive model, overestimates by 10 times the thickness of the viscous boundary layer given in *Equation 12*. Our choice was made because there is no scale separation between our topographic cues and the boundary layer thickness and drawing the shape of the top boundary of the diffusive region is a very difficult task. Consistently, the value of $|\nabla h|$ on the disk surface in our simulations is 10 times smaller of the estimation $|\nabla h|_0$ of the same quantity in our experiments reported in the manuscript. Note also, that as in all diffusive problems, the humidity gradient on any point of the bottom boundary (i.e. on the clay surface) depends on the distance of that point from the top boundary and on the topography. Thus, in principle, the size of the simulation box does not only affect the overall magnitude of the humidity gradient but also its shape. However, one observes that in our simulations the topographic cues are only 30% closer to the top boundary compared to the flat, bottom, surface, but the local gradient is 10–20 times larger. This evidence suggests that the 'curvature' effect is much more important than the 'distance' effect, and supports the fact that our approximation does not affect in a significant way the estimation of the relative importance of the humidity gradient at the bottom surface. Conversely, the approximation clearly affects the absolute value of the humidity gradient, but this can be easily 're-scaled' using the reference value $|\nabla h|_0$ which is defined in Materials and methods.

### Relationship between local surface curvature and evaporation field

Our chemical garden experiments, alongside with the stationary humidity field simulations described above both illustrate a relationship between substrate curvature and evaporation. Below, we provide theoretical arguments in support for the relation between evaporation and curvature. Here, we re-derive the expression for the humidity gradient at the surface of isolated spherical droplets of a given radius. Subsequently, in Appendix 3, we provide arguments demonstrating that similar results also apply with good approximation to our experimental substrate made available to termites.

### Diffusive evaporation from a spherical surface with a given curvature

As stated in the manuscript, the relationship between surface curvature and evaporation flux was observed by *Langmuir., 1918* already more than a century ago. Langmuir studied isolated droplets of an evaporating fluid and showed that the evaporation flux from an isolated droplet of radius $R$ is proportional to $R$, while the humidity gradient scales as $1/R$ at the droplet surface. For the sake of clarity this result is re-derived below following *Fuchs, 1947*. Let $h(r)$ be the humidity field around the droplet, at the equilibrium the diffusion equation in spherical coordinates reads:

$$0 = \frac{1}{r^2} \frac{\partial}{\partial r} \left( r^2 \frac{\partial h}{\partial r} \right) + \frac{1}{r^2 \sin\theta} \frac{\partial}{\partial \theta} \left( \sin\theta \frac{\partial h}{\partial \theta} \right) + \frac{1}{r^2 \sin\phi^2} \frac{\partial^2 h}{\partial \phi^2}.$$

(2)

As the system has spherical symmetry, $\partial_\theta = 0$ and $\partial_\phi = 0$ thus diffusion equation reduces to:

$$0 = \frac{\partial}{\partial r} \left( r^2 \frac{\partial h}{\partial r} \right),$$

(3)

which means that humidity gradient $\partial h / \partial r$ is $\partial h / \partial r = B/r^2$ where $B$ is a constant, and the humidity field must have the form:

$$h(r) = A - \frac{B}{r}$$

(4)

where $A$ is another constant. We assume then, that diffusion acts in a spherical shell between an inner sphere of radius equal to the radius of the droplet, and an outer sphere of radius $R_{out}$. Let $h(R) = h_0$ and $h(R_{out}) = h_{out}$; the constants $A$ and $B$ can then be obtained by substituting in 4 and they read:

$$A = h_0 - \frac{(h_0 - h_{out})R_{out}}{R_{out} - R} \qquad B = \frac{(h_{out} - h_0)RR_{out}}{R_{out} - R}.$$

(5)

We can then compute the humidity gradient at the droplet surface as:

$$\frac{\partial h}{\partial r}\bigg|_R = \frac{(h_{out} - h_0)R_{out}}{R(R_{out} - R)},$$ (6)

now if we assume that $R_{out} \gg R$ one sees that the humidity gradient at the droplet surface scales like $1/R$ and that the humidity flux evaporating from the droplet must scale like $R$.

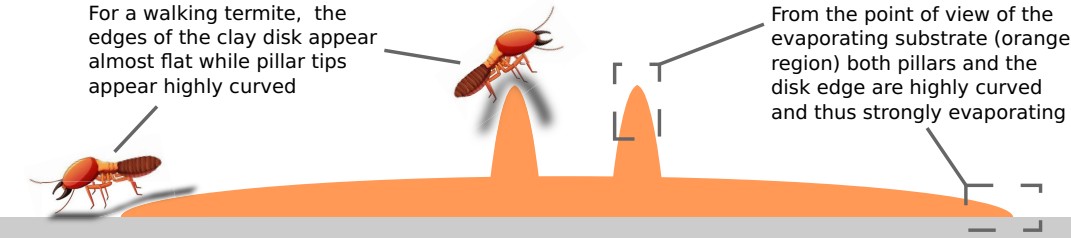

For a walking termite, the edges of the clay disk appear almost flat while pillar tips appear highly curved

From the point of view of the evaporating substrate (orange region) both pillars and the disk edge are highly curved and thus strongly evaporating

**Appendix 2—figure 1.** Sketch of the experimental setup showing the contrast between how sharp is the shape of the wet substrate (orange region) at the disk edge, and how flat the same region can appear to a walking termite.

# Appendix 3

## Scaling arguments for a quasi-stationary diffusive regime

Our experimental system is more complex than isolated droplets because the evaporating substrate is a porous medium, and its geometry is more convoluted than an isolated sphere. However, the same conclusions are valid if two important conditions are fulfilled which are:

1. We restrict our attention to the viscous boundary layer;
2. The porous medium is equally soaked everywhere.

As we showed in *Figure 1—figure supplement 2*, evaporation implies both humidity and temperature variations which can cause the onset of the so-called 'moist' convection, which is a complex phenomenon still under study by fluid-dynamicists *Vallis et al., 2019*. However, within the viscous boundary layer the system is dominated by diffusion. Below we show that the thickness of the viscous boundary layer is comparable to the thickness of termites, so fulfilling condition 1. Subsequently, we show that condition 2 is also fulfilled. In fact, by combining Washburn's results *Washburn, 1921* for the dynamics of capillary rise in a tube and an effective value for the size of pores in our clay disk *Diamond, 1970*, we can establish that there is a strong time scale separation between capillary rise through the clay disk and evaporation from the disk surface: evaporation is not limited by the capillary soaking of the substrate (see at the end of this section).

## Estimation of the thickness of the viscous boundary layer

Previous experiments by *Soar et al., 2019* indicated that termites live in the viscous boundary layer, that is, the thickness of the boundary layer is comparable with the typical body thickness of termites. Below we show that this assumption is justified also for our experiments with scaling arguments.

Traditionally, the thickness $\delta$ of the viscous boundary layer over a plate is estimated *Acheson, 1990* as:

$$\delta \propto \sqrt{\frac{l\nu}{v}}, \tag{7}$$

where $l$ and $v$ are typical scales for the size of the system and the speed of the fluid and $\nu$ is the kinematic viscosity of the fluid. This scaling was successfully used to model water evaporation flux from a vessel *Hisatake et al., 1993*. At 25 °C viscosity does not change much with relative humidity so we can take the value for dry air which is $\nu \sim 1.5 \cdot 10^{-5} \, \mathrm{m^2 s^{-1}}$ and we consider $l \sim 5 \cdot 10^{-2} \, \mathrm{m}$ which is the diameter of the clay disk. Estimating the velocity scale $v$ is a less easy task because, unlike in previous experiments by other authors (*Hisatake et al., 1993*; *Soar et al., 2019*), we are not imposing any air flow velocity in our experiments. However, this does not mean that the velocity is zero. In fact, above the viscous boundary layer humid, lighter air will flow upward pushed by the buoyancy force forming an uprising convective plume. By mass conservation, fresh dry air must come from the side of the disk with velocity which is parallel to the horizontal and comparable in magnitude with the velocity of the uprising plume. Air density is also affected by temperature, and as we showed in *Figure 1—figure supplement 2*, there is a temperature drop at the clay disk surface because of evaporation which causes air density to increase and thus slow down or inhibit convection. For the sake of simplicity, we then neglect temperature effects at this stage (though we will discuss them at the end of this section) which will give us an upper bound for $v$ and a lower bound for $\delta$.

To assess the buoyancy force we must quantify the density of humid air $\rho$. This can be estimated treating it as a mixture of perfect gases (Dalton's law) which leads to:

$$\rho = \rho_d - P_{sat}\phi\frac{R_d - R_v}{R_d R_v T}, \tag{8}$$

where subscripts $d$ and $v$ indicate dry air and water vapor respectively, $R$ is the specific gas constant, $T$ is the absolute temperature, $\Phi$ is the relative humidity and $P_{sat}$ is the vapor saturation pressure. Using Teten's formula for computing $P_{sat} \sim 3.2 \cdot 10^3 \, \mathrm{Pa}$ at 25 °C, the density variation $\Delta\rho_v$ between the surface of the clay disk ($\Phi = 100\%$) and far from it ($\Phi \sim 70\%$) reads:

$$\Delta\rho_v \sim 4.2 \cdot 10^{-3} \, \mathrm{kg m^{-3}}. \tag{9}$$

Using this value, we can estimate the Rayleigh number for our setup, which is:

$$Ra = \frac{g\delta_v H^3}{D_v \eta_a} \sim 3 \cdot 10^6,$$ (10)

where $D_v$ is the mass diffusivity of water vapor in air, $\eta_a$ is the dynamic viscosity of air and the length $H$ is the height of the column of air between our experimental setup and the ceiling of the room. One observes that $Ra$ is much larger than the critical Rayleigh number for convection $Ra_c \sim 10^3$, thus if temperature effects are neglected, air is certainly convecting. We can then estimate the velocity induced by buoyancy copying the choice of *Malkus and Thayer, 1964* for thermal convection:

$$v \sim \sqrt{gH\Delta\rho_v/\rho_d} \sim 0.25 \, \text{ms}^{-1},$$ (11)

which substituting in *Equation 7* gives

$$\delta \sim 2 \, \text{mm},$$ (12)

which is comparable with the body thickness of our termites ($\sim 1 \, \text{mm}$).

## Temperature effects

Below we discuss the effect of evaporation cooling and we estimate the density difference caused by the temperature drop at the clay disk surface. According to *Figure 1—figure supplement 2*, the temperature on the clay disk can be estimated to drop by about 1 °C, while the expansion coefficient of air at 25 °C is $\alpha \sim 3.4 \cdot 10^{-3} \, \text{kgm}^{-3}$ thus the density difference induced by the temperature drop is

$$\Delta\rho_T = \rho_d \alpha \Delta T \sim 4.0 \cdot 10^{-3} \, \text{kgm}^{-3},$$

which is very close to the value we find for the density difference induced by humidity (*Equation 9*). Note that this is a positive variation, i.e. cooler air is heavier, which should contrast and possibly inhibit convection. However, one should notice that if convection does not set, $\delta$ will be larger and the evaporation flux smaller as shown by *Equation 15*. This means that, even if the cooling caused by evaporation slows down or possibly stops the evaporation from time to time, the process might start again following a sort of intermittent sequence. In any case, temperature effects can only weaken convection and thus increase the thickness of the the viscous boundary layer $\delta$, in which case the conclusions of our paper are nothing but reinforced.

## Scaling arguments for a quasi-stationary diffusive regime

In the manuscript we stated that the clay disk is constantly replenished in water by capillary rise while evaporating from the surface. In particular, we assume that this is a quasi stationary process and that the value of moisture at the disk surface is approximately constant and air is saturated with vapor. This assumption relies on a strong scale separation between the typical time of evaporation and the time that water needs to rise up to the disk surface. Below we show that such a scale separation does exist in our experiments.

It is well known that a fluid can flow through a capillary up to a equilibrium height $H_c$ which is given by the following formula:

$$H_c = \frac{2\gamma \cos\theta}{\rho g r}$$ (13)

with $\gamma$ the surface tension, $\theta$ the contact angle, $\rho$ the fluid density, $g$ the gravity and $r$ the capillary diameter. The case of a porous medium is more complex but *Equation 13* can still be used choosing the adequate *effective* pore radius $r$, and we take $r \approx 10^{-7} \, \text{m}$ (*Diamond, 1970*). Clay is hydrophilic, which means that we can approximate $\cos\theta \approx 1$ and we remind that surface tension for water at 25 °C is $\gamma \approx 7 \cdot 10^{-2} \, \text{Nm}^{-1}\text{s}^{-1}$ and $\rho g \approx 10^4 \, \text{Nm}^{-3}$. One can then estimate $H_c$ as $H_c \approx 100 \, \text{m}$ in clay. One will notice that in our experimental setup water must climb no more than 6 mm from the reservoir to the top of our topographic cues thus the mechanism of capillary rise appear more than effective to constantly replenish the clay disk with water and gravity can be neglected in the rising dynamics. With such hypothesis *Washburn, 1921* showed that the capillary rise $h$ evolves with a diffusion law $h^2 = D_\gamma t$ where the effective diffusion coefficient $D_\gamma$ can be written as:

$$D_\gamma = \frac{\gamma \cos\theta r}{2\eta_w} \approx 4 \cdot 10^{-6} \, \text{m}^2\text{s}^{-1},$$ (14)

where $\eta_w = 10^{-3}\,\mathrm{Nm^2s^{-1}}$ is the dynamic viscosity of water. As we stated in the manuscript, close to the surface the uptake of water also happens by diffusion. The time scale separation between capillary rise and evaporation can then be estimated computing the corresponding diffusive flux $q_\gamma$ and $q_{ev}$ across the clay disk thickness $d$ and the viscous boundary layer $\delta$ respectively:

$$q_\gamma = \frac{SD_\gamma \rho_l}{d}, \qquad q_{ev} = \frac{SD_v \Delta \rho_v}{\delta}. \tag{15}$$

Approximating the surface $S$ to be the same for the two processes and taking $d = 6\,\mathrm{mm}$ as an upper boundary for the clay disk thickness (i.e. the height of the topographic cues), the ratio between the two fluxes reads:

$$\frac{q_\gamma}{q_{ev}} \sim \frac{D_v}{D_\gamma} \frac{\Delta \rho_v}{\rho_l} \frac{1}{3} \sim 10^3. \tag{16}$$

where we use the value of $\delta$ and $\Delta \rho_v$ obtained in section 8, the effective diffusion $D_\gamma$ from **Equation 14** and $D_v = 2.6 \cdot 10^{-5}\,\mathrm{ms^{-2}}$ the mass diffusivity of vapor at 25 °C (**Cussler, 1997**). One observes that time scales of capillary rise and evaporation are separated by three order of magnitude which means that our assumption is highly reliable. As an example it would remain valid even if we were about to commit an error of 2 order of magnitude in (over) estimating $r$ the effective pore size of our clay, which is by far the highest source of uncertainty in this estimation. Finally, note that the time scale of capillary rise is a decreasing function of the pore size $r$ (see **Equation 14**), while $H_C$ scales like $1/r$ (see **Equation 13**), thus our assumptions remain valid even if the new construction made by termites included bigger pores up to a size of 1 mm.

