## [Editor Report · eLife assessment]

This **valuable** study investigates the environmental drivers behind termite construction, focusing, in particular, on pellet deposition behavior, with the conclusion that termites likely sense curvature indirectly through substrate evaporation. The findings reconcile discrepancies between previous studies through experimental and computational approaches. While the strength of the evidence supporting these claims is **compelling**, the authors do not discuss how their results affect our understanding of insect nest construction or animal-built structures more broadly.

---

## [Referee Report · Joint Public Review]

In this manuscript the authors performed experiments and simulations which showed that substrate evaporation is the main driver of early construction in termites. Additionally, these experiments and simulations were designed taking into account several different works, so that the current results shine a light on how substrate evaporation is a sufficient descriptor of most of the results seen previously.

Through simulations and ingenious experiments the authors have shown how curvature is extremely correlated with evaporation, and therefore, how results coming from these 2 environmental factors can be explained through evaporation alone. The authors have continued to use their expertise of numerical simulations and a previously developed model for termite construction, to highlight and verify their findings. On my first pass of the manuscript I felt the authors were missing an experiment: an array of humidity probes to measure evaporation in the three spatial dimensions and over time. Technologically such an experiment is not out of reach, but the author's alternative (a substrate made with a saline solution and later measuring the salt deposits on the surface) was a very ingenious low tech solution to the problem.

The authors agree that future experiments should tackle finely controlled humidity levels and curvature in order to have a more quantitative measure termite behaviour, but the work done so far is more than sufficient to justify their current claims.

In the revised text, the authors have added more clarity into different biological systems in which these results could be applied. Perhaps what it would have been beneficial to also add more information on how the resulting algorithms of constructions can be used in swarm robotics with collective construction, both macro and micro, but I acknowledge that the style of the paper does focus more on the biological aspects

The results presented here are so far the best attempt on characterizing multiple cues that induce termite construction activity, and that possibly unifies the different hypothesis presented in the last 8 years into a single factor, resulting into a valuable addition to the field. More importantly, even if these results come from different species of termites than some of the previous works, they are relatable and seem to be mostly consistent, improving the strength of the author's claims.

---

## [Author Response]

The following is the authors’ response to the previous reviews.

This paper now provides a convincing presentation of valuable results of the drivers of nest construction for one termite species, and they briefly discuss possible relevance to other termite species. However, the authors have not yet addressed how their results may be important outside the field of termite nest construction. I could imagine the significance of the paper being elevated to important if there is a broader discussion about the impact of this work, e.g., the relevance of the results, the approach, and/or next steps to related fields outside of termite nest construction.

Reading our manuscript again, we have to agree with the reviewer that we mostly focused the discussion of our results in the context of termite construction, without attempting to generalise to other systems. To some extent we still defend this choice, as we prefer not to make too many claims on the relevance of our results beyond what we can reasonably support with our own experimental results. However, we thought that it would be appropriate – as suggested by the reviewer – to add at least one paragraph to indicate how our results could be extrapolated to other systems. This new paragraph is now at the end of the discussion section.

Here we elaborate a bit further on this point: first of all, while termites certainly build the most complex structures found in the natural world, there aren’t many other animals that are capable of collectively building complex structures. Typically, collective building activity is limited to highly social (typically eusocial) animals, but other social insects, such as ants and wasps, are phylogenetically distant from termites, their nests are often different (the large majority of ant nests only comprise excavated galleries with little construction, while wasp nests tend to comprise multiple repeated patterns that could be produced from stereotyped individual behaviour). Because of these differences, drawing a comparison between the mechanisms that regulate termite architecture and those that regulate other forms of animal architecture would be too speculative. One domain, however, where similar mechanisms to those that we describe here could operate is that of pattern formation at the cellular and tissue level, where surface curvature was shown to drive different phenomena from cell migration to tissue growth. A comment on this is now added in the manuscript at the very end of the discussion.

Similarly, on a related note, as someone not directly in the field of termite nest construction but wanting to understand the system (and the results) presented here in a broader context, I found the additional information about species and natural habitat very helpful and interesting, though I was rather disappointed to find it relegated to supplementary material where most readers will not see it.

We considered this suggestion to present more information about the natural nesting habits of the termites that we study into the main text, and we moved that part to the Materials and methods. We feel that the nesting habits of the termites that we studied here are not too central to the problem that we want to focus on, of how they coordinate their building activity. In fact, there is a large variety of nesting habits across termite genera and species, but we believe that, at a basic level, the mechanisms that we describe here would also apply to species with different nesting habits, because our results are consistent with what is described in the scientific literature for other termite species. As our introduction is already a bit long, we left this description of Coptotermes nesting habits in the Materials and methods, where, hopefully, it will still be accessible and useful to readers interested in finding this information.

When providing responses to reviewers, please directly address the reviewers’ comments point-by-point rather than summarizing comments and responding to summaries.

We apologize for our previous way to respond to comments and thanks the reviewer for his remark as we learn to navigate through the eLife reviewing system (where some comments are repeated in the overall assessment and in the feed-back of individual reviewers).

Figure 2 colors: Panels A and E and maybe B do not seem colorblind-friendly. I suggest modifying the colormaps to address this.

We have changed the colormaps of figures A,B and E which are now colorblind-friendly.

Line 180: This system is not in equilibrium. Perhaps the authors mean "steady-state?" I suggest reviewing language to ensure that the correct technical terms are used.

We have now corrected this.